# Remove Artifacts from a Single-Channel EEG Based on VMD and SOBI

**DOI:** 10.3390/s22176698

**Published:** 2022-09-04

**Authors:** Changrui Liu, Chaozhu Zhang

**Affiliations:** School of Information and Automation Engineering, Qilu University of Technology (Shandong Academy of Sciences), Jinan 250353, China

**Keywords:** EEG, artifact removal, variational mode decomposition, second order blind identification, fuzzy entropy

## Abstract

With the development of portable EEG acquisition systems, the collected EEG has gradually changed from being multi-channel to few-channel or single-channel, thus the removal of single-channel EEG signal artifacts is extremely significant. For the artifact removal of single-channel EEG signals, the current mainstream method is generally a combination of the decomposition method and the blind source separation (BSS) method. Between them, a combination of empirical mode decomposition (EMD) and its derivative methods and ICA has been used in single-channel EEG artifact removal. However, EMD is prone to modal mixing and it has no relevant theoretical basis, thus it is not as good as variational modal decomposition (VMD) in terms of the decomposition effect. In the ICA algorithm, the implementation method based on high-order statistics is widely used, but it is not as effective as the implementation method based on second order statistics in processing EMG artifacts. Therefore, aiming at the main artifacts in single-channel EEG signals, including EOG and EMG artifacts, this paper proposed a method of artifact removal combining variational mode decomposition (VMD) and second order blind identification (SOBI). Semi-simulation experiments show that, compared with the existing EEMD-SOBI method, this method has a better removal effect on EOG and EMG artifacts, and can preserve useful information to the greatest extent.

## 1. Introduction

Electroencephalogram (EEG) is an electrical signal behavior caused by brain activity that can reflect the electrophysiological activities of brain nerve cells and can obtain a large amount of physiological or psychological activities and some disease information. It is one of the effective means to detect brain function and condition. However, in the acquisition process, due to the influence of the acquisition equipment, the external environment, and human physiological activities, some artifacts [1] will inevitably be mixed in, such as EOG artifacts, EMG artifacts, etc. These artifacts will have a great impact on the processing of subsequent EEG signals; thus, we need to develop some effective methods to remove these artifacts and to preserve useful information to the greatest extent.

In the early stage, the research object is mostly the artifact removal of multi-channel EEG signals, mainly including regression and filtering methods [2,3]. These two methods require reference channel signals. If there is no available reference signal, BSS is the most commonly used method, and research shows that the ICA algorithm is more suitable for removing EOG artifacts [4,5]. At the same time, wavelet transform is also very effective in removing EOG artifacts [6]. Compared with EOG artifacts, it is more difficult to remove EMG artifacts, on the one hand, because EMG lacks reference signals, and on the other hand, it is characterized by a high amplitude, a wide frequency band, and mixing with EEG [7]. Therefore, some researchers found that the ICA algorithm could not effectively remove EMG artifacts from EEG signals [8,9]. In recent years, canonical correlation analysis (CCA) [9,10] and independent vector analysis (IVA) [7] have been used to remove EMG artifacts and have demonstrated their superior performance to ICA in certain aspects; however, the ICA algorithm is still recognized as the most widely used blind source separation algorithm [11]. At present, although there is no specific method that can effectively remove artifacts from all types of EEG signals; the BSS method is still the most extensive and effective method for processing EEG signals. Similar to other methods, BSS also has its limitations. This method requires the number of channels of the source signal and a small number of channels will reduce the separation effect [12]. Therefore, it is generally not directly used for the signal with few channels or a single channel. Nowadays, with the development of portable EEG acquisition systems, in many cases, only few-channel or even single-channel EEG can be collected. Especially in biomedicine, portable EEG acquisition systems will be more suitable for social needs. In order to overcome the limitation that BSS cannot be directly applied to single-channel EEG, researchers usually introduce a signal decomposition method, that is, the original EEG is decomposed into multiple signal components as the input of BSS, then the artifacts are identified and removed, and finally the pure EEG is reconstructed. Mijovic [13] combined EMD with ICA, first performed EMD decomposition on EEG, and used the generated eigenmode function as the input of ICA to successfully separate the artifact components in a single-guided EEG. However, signals decomposed by the EMD method will have modal mixing phenomenon, resulting in the incomplete removal of artifacts or the removal of useful information by mistake. Subsequently, its improved algorithm EEMD was proposed [14], and the literature [15] proposed the method of combining EEMD and BSS to remove artifacts in a single-channel EEG, however residual noise and other phenomena may still occur [16]. The literature [17] proposed a method combining singular spectrum analysis (SSA) and ICA to deal with the problem of multiple artifacts and achieved certain results. Generally speaking, most of the current methods are for the removal of a single artifact, but there are few studies on the removal methods for solving multiple artifacts at the same time.

Aiming at the above problems, this paper proposes a method combining variational mode decomposition (VMD) and the second order blind identification (SOBI) algorithm. It not only solves the modal aliasing phenomenon of the EMD decomposition method, but also overcomes the limitations of the ICA algorithm for dealing with different artifacts. There are three main steps in this method. First, single-channel EEG signals are decomposed into multi-channel datasets by the VMD method after optimizing parameters. VMD is proposed on the basis of the EMD method. Compared with EMD and its derived improved algorithms, VMD solves the problem of modal mixing, effectively suppresses end-point effects, and exhibits excellent noise robustness in practical applications. VMD methods have been used to analyze EEG artifact removal. Chinmayee Dora [18] proposed a method based on VMD. Before signal decomposition, multi-scale sample entropy was used to identify EEG channels mixed with EOG artifacts, and then VMD decomposition was used to identify and remove artifacts from the obtained IMF (intrinsic modal function) and to reconstruct a pure signal. A number of studies have shown that the VMD method is better than the EMD method in terms of the decomposition effect, but it has many parameters and needs to be specifically analyzed for different input signals.

The second step is to use the BSS method to process the decomposed multi-channel data. The SOBI algorithm in the ICA method is selected here. Although the ICA method is somewhat controversial, it is still the main solution for EEG artifact removal. ICA is divided into several different implementations. This paper chooses the SOBI algorithm. SOBI is a method based on second-order statistics (SOS). The use of covariance matrices for joint approximate diagonalization to achieve the blind source separation of observed signals has appeared in some EEG artifact removal studies [19]. Jose [20] pointed out that the SOBI algorithm based on second-order statistics is better than other ICA methods based on higher-order statistics in dealing with different kinds of artifacts.

The third step is to identify the artifact components. The signal components processed by the above VMD–SOBI method have theoretically separated the EEG signal and the artifact signal, and now the artifact signal needs to be identified. Although the individual components can be distinguished by human observation of the time domain and the frequency domain diagrams of each signal component, the artifact components are identified here by calculating the fuzzy entropy value of each signal component.

The structure of the paper is as follows: Section 2 introduces the method used in this paper. Section 3 provides the simulation results and discusses the analysis. Section 4 analyzes the influence of parameters in the VMD method and proposes an optimization method for the most important parameters among them; this section is a supplement to the above methods. Finally, the full text work is summarized.

## 2. Methods

### 2.1. The Basic Idea of Single-Channel EEG Signal Artifact Removal

BSS is the most widely used method in multi-channel EEG signal artifact removal methods. For single-channel EEG signals, BSS is generally not used directly. An algorithm such as ICA is more dependent on the number of channels. The more channels, the better its performance. In recent years, the idea of solving single-channel EEG signals is to decompose the signal first, and then separate the signal from the artifact by means of BSS. Then, the artifact components are identified by the identification method, and finally the pure EEG signal is reconstructed. The following Figure 1 is the flow chart of the artifact removal of a single-channel EEG signal.

### 2.2. VMD

Variational mode decomposition (VMD) [21] was proposed by Dragomiretskiy and Zosso on the basis of empirical mode decomposition (EMD), which is similar to EMD results. VMD redefines the intrinsic modal function IMF as the bandwidth-constrained AM-FM signal.
(1)$$u_{k}\left(t\right)=A_{k}\left(t\right)\cos\left(\phi_{k}\left(t\right)\right)$$
where $A_{k}\left(t\right)$ is the amplitude envelope of $u_{k}\left(t\right)$ and $A_{k}\left(t\right)\geq 0$, $\phi_{k}\left(t\right)$ is the instantaneous phase of $u_{k}\left(t\right)$, and $\phi_{k}^{\prime }\left(t\right)\geq 0$.

VMD transforms the signal decomposition process into a variational framework and realizes the adaptive decomposition of the signal by searching for the optimal solution of the constrained variational model. During the iterative solution of the variational model, the frequency center and the bandwidth of each component of the IMF are continuously updated. Finally, the adaptive division of the signal frequency band is completed according to the frequency characteristic of the signal and a plurality of narrow-band IMF components are obtained. Assuming that the original signal is decomposed into K IMF components through VMD, the expression of the constrained variational model is
(2)$$\left\{\begin{array}{c} \underset{\left\{u_{k}\right\},\left\{\omega_{k}\right\}}{\min}\left\{\underset{k}{\sum }∥\partial_{t}\left[\left(\delta \left(t\right)+\frac{j}{\pi t}\right)*u_{k}\left(t\right)\right]e^{-j\omega_{k}t}{∥}_{2}^{2}\right\}\\ s.t.\overset{K}{\underset{k=1}{\sum }}u_{k}=f \end{array}\right\}$$
where $\left\{u_{k}\right\}$ = $\left\{u_{1},u_{2},\ldots ,u_{k}\right\}$ is the K IMF components decomposed by the VMD method, $\left\{\omega_{k}\right\}$ = $\left\{\omega_{1},\omega_{2},\ldots ,\omega_{k}\right\}$ is the frequency center of each IMF component $u_{k}$, $\partial_{t}$ is the partial derivative of the function time, δ(t) is the unit impulse function, j is the imaginary unit, ∗ means convolution, and f represents the original signal. The quadratic penalty function term α and the Lagrangian multiplication operator λ are introduced to solve the optimal solution of the above constrained variational problem, and the expression is:(3)$$L\left(\left\{u_{k}\right\},\left\{\omega_{k}\right\},\lambda \right)\;=\alpha \underset{k}{\sum }∥\partial_{t}\left[\left(\delta \left(t\right)+\frac{j}{\pi t}\right)*u_{k}\left(t\right)\right]e^{-j\omega_{k}t}{∥}_{2}^{2}+∥f-\underset{k}{\sum }u_{k}{∥}_{2}^{2}+\langle \lambda ,f-\underset{k}{\sum }u_{k}\rangle$$

In the formula, α is the quadratic penalty factor. The function is to reduce the interference of Gaussian signals. In order to ensure the accuracy of signal reconstruction, α is generally set to be a large enough positive number, and λ is the Lagrange multiplication operator. The optimal solution of the constrained variational model of Equation (2) is to use the alternating direction multiplier algorithm to obtain the saddle point of the Lagrangian function, thereby obtaining the narrow-band IMF component. The specific process is as follows:
(1)Initialization parameter $\left\{u_{k}^{1}\right\}$,$\;\left\{\omega_{k}^{1}\right\}$, $\lambda^{1}$,  n=0;(2)n=n+1;(3)k=k+1, traversing k=1−K, update ${\hat{u}}_{k}^{n+1}$ and $\omega_{k}^{n+1}$ with the following formulas, respectively:(4)$${\hat{u}}_{k}^{n+1}(\omega )=\frac{\hat{f}(\omega )-\frac{\sum_{i\neq k}{\hat{u}}_{i}(\omega )+\hat{\lambda }(\omega )}{2}}{1+2\alpha {\left(\omega -\omega_{k}\right)}^{2}}$$
(5)$$\omega_{k}^{n+1}=\frac{\int_{0}^{\infty }\omega {\left|{\hat{u}}_{k}^{n+1}(\omega )\right|}^{2}d\omega }{\int_{0}^{\infty }{\left|{\hat{u}}_{k}^{n+1}(\omega )\right|}^{2}d\omega };$$(4)Update the Lagrange multiplier λ
(6)$${\hat{\lambda }}^{n+1}\left(\omega \right)={\hat{\lambda }}^{n}\left(\omega \right)+\gamma \left(\hat{f}\left(\omega \right)-\sum_{k}{\hat{u}}_{k}^{n+1}\left(\omega \right)\right).$$


In the formula, γ  is the noise tolerance, which meets the fidelity requirements of signal decomposition. ${\hat{u}}_{k}^{n+1}\left(\omega \right)$, ${\hat{u}}_{i}\left(\omega \right)$, $\hat{f}\left(\omega \right)$, $\hat{\lambda }\left(\omega \right)$ corresponds to the Fourier transform of $u_{k}^{n+1}\left(t\right)$, $u_{k}^{n}\left(t\right)$, f(t), λ(t), respectively.
(5)Repeat steps 2–4 until the convergence condition of the following equation is satisfied
(7)$$\sum_{k}\frac{∥{\hat{u}}_{k}^{n+1}-{\hat{u}}_{k}^{n}{∥}_{2}^{2}}{∥{\hat{u}}_{k}^{n}{∥}_{2}^{2}}<\varepsilon .$$


For a given judgment accuracy ε>0, end the loop.

Compared with the recursive decomposition mode of EMD, VMD converts the signal decomposition into a variational decomposition mode, the essence of which is a more adaptive Wiener filter group. VMD can realize the adaptive segmentation of each component in the signal frequency domain; it can effectively overcome the mode aliasing phenomenon generated in EMD decomposition and has stronger noise robustness and weaker end effect than EMD.

### 2.3. SOBI

The blind source separation method adopted in this paper is the second order blind identification algorithm (SOBI) [22]. SOBI is a method to achieve signal blind source separation by joint approximate diagonalization of the covariance matrix. Different from the ICA algorithm, ICA adopts high-order statistics, SOBI adopts second-order statistics, and the separation effect is not dependent on whether the source signal obeys Gaussian distribution, so it is suitable for processing EMG artifacts in EEG signals.

The basic model of blind source separation is:(8)x(t)=As(t)+n(t)

In the formula, $s\left(t\right)={\left[s_{1}\left(t\right),s_{2}\left(t\right),\ldots \ldots ,s_{n}\left(t\right)\right]}^{T}$ is the source signal, $x\left(t\right)={\left[x_{1}\left(t\right),x\left(t\right),\ldots \ldots ,x_{m}\left(t\right)\right]}^{T}$ is the m-channel signal collected by the sensor, and A is the m×n-dimensional mixing matrix.

For the above model, the steps of SOBI are as follows:
(1)First, pre-whitening is performed on the observed signal x(t), and the whitening matrix Q is calculated, in order to remove the correlation between channels and to improve the decomposition effect. The whitened signal is denoted as z(t).
(9)z(t)=Qx(t)(2)Calculate the sampling covariance matrix of multiple delays $\tau \in \left\{\tau_{1},\tau_{2},\ldots \ldots \tau_{k}\right\}$ of z(t)
(10)$$R\left(\tau \right)=E\left[z\left(t+\tau \right)z{\left(t\right)}^{T}\right]=AR_{z}\left(\tau \right)A^{T}.$$(3)For each covariance matrix $R\left(\tau_{j}\right)$ calculated by the above formula, perform joint approximate diagonalization to calculate the orthogonal matrix V:(11)$$V^{T}R\left(\tau_{j}\right)V=D_{j}$$
where $\left\{D_{j}\right\}$ is a set of diagonal matrices.(4)Estimate the mixing matrix A and the source signal y(t):(12)$$A=Q^{-1}V$$
(13)$$y\left(t\right)=A^{-1}x\left(t\right)=V^{T}Qx\left(t\right).$$


### 2.4. Artifact Recognition

The signal components obtained after being processed by VMD and SOBI need to be further identified as artifact components and then reconstructed to obtain pure EEG signals. The concept of fuzzy entropy (FE) [23] is introduced here. Fuzzy entropy is a method to measure the complexity of a signal sequence. Generally speaking, the more complex the signal, the larger the entropy value, and vice versa. According to this feature, the artifact component can be identified by calculating FE, and the calculation formula is as follows:(14)$$FE=\underset{n\to \infty }{\lim}\left(ln\Phi^{m}\left(n,\gamma \right)-ln\Phi^{m+1}\left(n,\gamma \right)\right)$$

In the formula, $\Phi^{m}\left(n,\gamma \right)$ is the calculated similarity, m is the embedding dimension, n is the gradient of the number boundary, and γ is the similarity tolerance.

## 3. Experimental Simulation and Analysis

### 3.1. Experimental Data

Before experimental simulation, the simulation signal is first constructed. In order to verify the effect of the VMD method on the removal of artifacts in EEG signals, the simulation data X(t) used here are composed of formal pure EEG signals $X_{EEG}\left(t\right)$. and artifact signals, so the simulation data are as follows:$$X\left(t\right)=X_{EEG}\left(t\right)+\theta X_{EMG}\left(t\right).$$

In the formula, the signal-to-noise ratio (SNR) of the original mixed signal can be adjusted by changing the value of the parameter θ.

The pure EEG signals selected in this paper are obtained from the Graz Data Set B database of BCI Competition IV [24]. The database uses three electrodes to record EEG signals (C3, Cz, and C4) at a sampling frequency of 250 Hz. The recorded EEG data included a screen dynamic range of ±100 uV and a feedback segment dynamic range of ±50 uV. Digital signals are bandpass filtered at 0.5–60 Hz, and notch filters are provided at 50 Hz. The artifact signals selected were from the public dataset of the University of Twente [25]. The dataset consists of four-channel EMG and four-channel EOG, among which one-channel EMG data and one-channel EOG data are selected to construct simulation signals. All data are shown in Figure 2.

### 3.2. Selection of VMD Parameters

Before decomposing the VMD on the input signal, it is necessary to determine the parameters of the VMD. Combined with the simulation experiments under various parameters of VMD in Section 4, the settings of the VMD parameters in this paper are as follows: the number of decompositions K=3, the noise margin (fidelity parameter) is γ=0, the initial center frequency is $\left\{{\hat{\omega }}_{k}^{1}\right\}=\left\{5,30,150\right\}$, the quadratic penalty factor α = 1500, and the convergence parameter $\varepsilon =10^{-5}$.

### 3.3. VMD-SOBI

The mixed signals were decomposed by the VMD method after the parameters were determined. Then, the decomposed IMF signal component is used as the input of SOBI, and blind source separation is performed to further separate the EEG signal from the artifact, as shown in Figure 3.

Calculate the fuzzy entropy of each component, as shown in the following Table 1.

According to the fuzzy entropy index of each component, IC2 was identified as an EOG artifact, and IC3 was identified as an EMG artifact, so it is set to zero. The reconstructed signal is obtained by inversely transforming the remaining IC components, as shown in Figure 4.

In order to further verify the denoising effect of the combined method, the SNR, RRMSE, and CC parameters are used as performance indicators, and the effectiveness and superiority of the proposed method is proved by comparison with other methods. Other methods used here are the EEMD-SOBI and the EEMD-ICA methods. The following is the calculation formula of several performance indicators:SNR
(15)$$\eta =10\times lg\left(\frac{\sum_{i=1}^{n}x_{i}{}^{2}}{\sum_{i=1}^{n}{\left[Y_{i}-x_{i}\right]}^{2}}\right)$$
2.RRMSE
(16)$$\sigma =\sqrt{\frac{\sum_{i=1}^{n}{\left(Y_{i}-x_{i}\right)}^{2}}{\sum_{i=1}^{n}x_{i}{}^{2}}}$$
3.CC
(17)$$\gamma =\frac{\sum^{\;}\left(x-\bar{x}\right)\left(Y-\bar{Y}\right)}{\sum^{\;}{\left(x-\bar{x}\right)}^{2}\sum^{\;}{\left(Y-\bar{Y}\right)}^{2}}$$

The following Table 2 shows the respective performance indexes of different methods at the original SNR of −1 db.

The following Figure 5 shows the relative root mean square errors and correlation coefficients of the four methods at different SNR (−1.5~1.5 dB).

The above figure shows the performance comparison between EEMD-SOBI and VMD-SOBI. The lower the CC value, the worse the ability to retain the effective signal of the original signal; the higher the RRMSE value, the worse the ability to remove the artifacts. It can be seen that the VMD-SOBI method proposed in this paper is better than the existing EEMD-SOBI method in removing EOG and EMG artifacts in EEG signals. The energy to preserve useful information is also stronger, especially at low SNRs. Moreover, it can be seen that the decomposition ability of VMD is less affected by the SNR than EEMD, which proves the superiority of VMD-SOBI in processing EOG and EMG.

## 4. Experimental Parameter Problem

### 4.1. Influence of VMD Parameters on Results

Through the above simulation results, the superiority of the VMD method can be seen, and through the description of the principle and decomposition steps of the VMD method, it can be seen that several parameters need to be set in advance in the VMD. It mainly includes the number of modal decomposition K, the penalty factor α, the initial center frequency $\left\{{\hat{\omega }}_{k}^{1}\right\}$, the noise tolerance parameter γ of updated $\hat{\lambda }$, and the convergence condition parameter ε. These parameters will have different degrees of influence on the decomposition effect of VMD, thus the influence of the above parameters will be analyzed, respectively.

The VMD method used in this paper achieves the effect of decomposing the signal components by optimizing the optimal center frequency of each source signal through multiple iterations. Therefore, it can be judged whether the decomposition effect of VMD is effective according to the frequency range of each source signal in Table 3. If the decomposed signal components are not within the following ranges, it is considered to be modal mixing.

#### 4.1.1. Number of Modal Decomposition K

As can be seen from the derivation of VMD, the value of IMF K is critical to VMD decomposition. An inappropriate number of IMF may lead to the wrong center frequency $\left\{{\hat{\omega }}_{k}^{n}\right\}$, and the IMF signals surrounding the center frequency may differ from the actual signal, resulting in mode mixing. In general, when the value of K is too small, the decomposed IMF will contain multiple frequency bands. When the value of K is too large, a frequency band will decompose into several IMF components.

In order to further analyze the influence of the setting of the number of IMFs on the VMD decomposition effect, the VMD method with K=2, 5 is set to decompose the simulated signal X(t), respectively. In addition, other parameter variables of VMD should be fixed: α=1000, noise tolerance γ=0, convergence condition $\varepsilon =10^{-5},$ and the initial center frequency $\left\{{\hat{\omega }}_{k}^{1}\right\}$ of each IMF is set to 0.

The time domain diagram and frequency domain diagram of decomposition results of K=2 are shown in Figure 6 below. It can be seen that the EMG signal with the higher frequency has not been decomposed and it is under-decomposed at this time. Figure 7 shows the decomposition of the original signal by VMD with K=5 and its time domain and frequency domain diagrams are shown below. It can be seen that frequency aliasing appears in the IMF3, IMF4, and IMF5 components (The part circled in red.), indicating that the value of K is too large and that the phenomenon of over-decomposition occurs. Although for this simulation signal, the over-decomposed part is the artifact, which has no significant impact on the EEG signal, considering that the larger the K value is, the longer the running time is, a more appropriate K value should be found.

Because the number of decomposition is the most influential parameter in VMD, there are many studies on its selection. Some articles mentioned that the number of modal decomposition K value should be equal to the number of components of the simulation signal, that is, if the simulation signal has an n signal superposition, then the K value should be selected n (also said that n+1). Through the experimental simulation results, it is concluded that if the simulated signal is only a few sine and cosine signals superimposed on each other, then it is indeed appropriate to select the number of simulated signal components for the K value, because the independence between each source signal is strong. In fact, for the truly collected EEG signals, the independence between EEG signals and artifact signals is not so strong, so only associating the selection of *K* value with the number of independent source signals may lead to unsatisfactory decomposition effect. In addition, there is a very important problem that, in the actual situation, we do not know whether the mixed signal is composed of several source signals. These are issues that need to be considered.

#### 4.1.2. Noise Tolerance γ

It can be seen from the updated formula of the Lagrangian multiplier λ that ${\hat{\lambda }}_{k}$ is used as the frequency residual term of the VMD method after each iteration calculation, so the initial ${\hat{\lambda }}_{k}^{1}$ should be set to 0. The γ used to update ${\hat{\lambda }}_{k}$ is the weight parameter to retain the residual term, and γ is called the fidelity parameter (also called noise tolerance). The value range of γ is [0, 1]. Whether γ is 0 or 1 should be determined by analyzing the actual signal. In order to further analyze the influence of the parameter γ on the VMD decomposition results, the simulation signal is used to change the SNR, and the VMD with γ=0 and γ=1 is used to decompose the original simulation signal, respectively. The other parameters remain the same, specifically: K=3, α=1000, convergence condition $\varepsilon =10^{-5}$, and the initial center frequency of each IMF is set to 0. The final convergence center frequency and convergence times for simulated signals with different SNRs are shown in Figure 8 below.

It can be seen from the above Figure 8 that when the SNR range of the original signal is from −12 dB to 8 dB, the VMD when γ=0 can converge faster, thereby effectively extracting the center frequency and the effective signal around the center frequency. When γ=1, the number of iterations will always reach the maximum number of iterations n=500. In fact, the decomposition process has not reached the convergence result, so there is a certain deviation between the optimization of the center frequency and the actual center frequency. However, when the SNR of the original signal is relatively high (greater than 8 dB), the VMD when γ=1 can stably extract the actual center frequency (the frequency bands will alternate occasionally, which does not affect the final result), but when γ=0, the center frequency cannot be effectively extracted, the reason may be that the initial center frequency is set to search from 0 Hz, while the residual component of this layer is ignored in the layer-by-layer iteration of VMD when γ=0, resulting in the loss of more energy. When the IMF3 component iterates to the frequency of about 20 Hz, the convergence condition has been satisfied, resulting in the phenomenon of modal mixing, which makes the decomposition effect of VMD unsatisfactory. While the signal of γ=1 has high fidelity, so the frequency energy of the high frequency part can be preserved. Through the above simulation implementation, it can be concluded that when γ=1, the VMD has poor convergence ability when the SNR of the input signal is low, making the decomposition effect unsatisfactory, but it converges in the environment when the SNR of the input signal is high. The effect is better than γ=0. In fact, some studies show that when γ = 0, the decomposition problem of VMD with high SNR can be solved by presetting the initial center frequency close to the actual source signal frequency. Next, by changing the setting method of the initial center frequency to fixed, the original signal is simulated again. The parameters of VMD are as follows: K=3, α=1000, convergence condition $\varepsilon =10^{-5}$, and the initial value of each IMF center frequency $\left\{{\hat{\omega }}_{k}^{1}\right\}=\left\{5,30,150\right\}$ (the initial center frequency here is set according to the frequency range of each source signal). The final convergence center frequency and convergence times for simulated signals with different SNRs are shown in Figure 9 below. It can be seen that the VMD when γ=1 still has a poor decomposition effect when the SNR is low, and the effect is better when the SNR is high. When γ=0, the VMD can stably extract the actual center frequency in the entire range of SNR, which solves the problem that the decomposition effect is not good at a higher SNR when the initial center frequencies are all set to 0. Therefore, in this paper, the advantage of the VMD noise tolerance parameter γ=0 for EEG signal decomposition is obvious, so the parameter γ is set to 0 in the following VMD methods.

#### 4.1.3. Initial Center Frequency

The IMF component $\left\{{\hat{u}}_{k}^{n+1}\right\}$ decomposed by VMD is updated according to the center frequency $\left\{{\hat{\omega }}_{k}^{n}\right\}$ of the previous layer, so setting the accurate center frequency in advance can more stably and effectively achieve the best center of each IMF. It can also reduce the number of optimizations of $\left\{{\hat{\omega }}_{k}^{n}\right\}$ and $\left\{{\hat{u}}_{k}\right\}$ to improve the iterative efficiency of VMD. In order to further analyze the influence of the initial center frequency on the VMD decomposition results through the simulated signal, two groups of simulated signals with SNR of −6 dB and 6 dB were constructed, and three initial center frequency setting schemes were defined: the first is the initial center frequency that is all set to ${\left\{{\hat{\omega }}_{k}^{1}\right\}}_{1}=0$; the second is the random initial center frequency ${\left\{{\hat{\omega }}_{k}^{1}\right\}}_{2}=r_{i}$, where $r_{i}$ is a random number within the analysis frequency range; the third is a fixed initial center frequency, where the initial center frequency of each IMF is set as ${\left\{{\hat{\omega }}_{k}^{1}\right\}}_{3}=\left\{5,30,150\right\}$. By setting a penalty factor of 100 to 3000, the center frequency convergence of different initial center frequency settings is analyzed, and the result is shown in Figure 10.

As can be seen from the Figure 10, in the simulated signal with a SNR of −6 dB, the three initial center frequency settings can basically effectively extract the frequency of the original signal, there is a small amount of aliasing in the IMF components of the VMD decomposition when the initial center frequency is set to zero and random when the penalty factor is small or large. However, when the SNR increases, as shown in Figure 11, only ${\left\{{\hat{\omega }}_{k}^{1}\right\}}_{3}$, that is, VMD with a fixed initial center frequency, can effectively decompose the input signal within the range of the penalty factor, the fault tolerance of VMD to the penalty factor under the other two settings decreases with the increase of SNR. It can be seen that the effective penalty factor range is only about 200 to 500 in the simulated signal with a SNR of 6 dB. It can be concluded that, compared with these two cases, the VMD whose initial center frequency is set to be fixed is less affected by the penalty factor and the SNR, and the robustness of the algorithm is better.

#### 4.1.4. Quadratic Penalty Factor α

The quadratic penalty factor α can be thought of as the standard deviation of the white noise added in the Wiener filter. Some studies have found that the value of α will affect the search bandwidth range of each IMF in the actual decomposition process. When the value of α is smaller, the search range of $\left\{{\hat{u}}_{k}\right\}$ around the center frequency $\left\{{\hat{\omega }}_{k}\right\}$ is larger and the bandwidth of each IMF component obtained by decomposition is larger. Conversely, when the value of α is larger, the search range is smaller and the bandwidth of the IMF is smaller. In order to specifically analyze the influence of the quadratic penalty factor on the decomposition effect of the VMD method, the simulated signal is used to change the value of the penalty factor and the input signal is decomposed by VMD under different SNRs (−6 dB, 1 dB). Other parameters remain the same and the specific settings are: K=3, γ=0, convergence condition $\varepsilon =10^{-5}$, and initial center frequency of each IMF $\left\{{\hat{\omega }}_{k}^{1}\right\}=\left\{5,30,150\right\}$. The final convergence frequency and convergence times of the simulated signals under different SNRs are shown in Figure 12 below. It can be seen that the convergence times of the simulated signals under different SNRs are lower when the penalty factor is lower (below 500), the convergence times are larger, and the corresponding decomposition efficiency becomes lower. After that, with the increase of the penalty factor, the number of iterations has been kept at a low level, that is to say, after the value of the penalty factor is 500, the increase of its value will not affect the decomposition rate of VMD. However, whether the decomposition results have any influence needs further experimental verification.

In order to further analyze the influence of different penalty factors on the decomposition results, the following comparative experiments were designed: a set of simulated signals with SNRs of −6 dB and 1 dB were constructed, the penalty factor ranges from 100 to 5000, and the step size is 100. Other parameter settings are the same as the VMD parameter settings in this section. The influence of the penalty factor is analyzed by calculating the SNR between the reconstructed EEG signal and the original pure EEG signal in the VMD decomposition results of the simulated signals with different SNRs.

As shown in Figure 13, through the simulation results of the input signal decomposition of different SNRs by VMD under different penalty factors, as long as the penalty factor is not too small (below 500), the decomposition results and efficiency of VMD will not be too bad. However, the optimal range of the penalty factor is also different for input signals with a different SNR. With the increase of the SNR of the input signal, the value of the optimal range also decreases gradually (for example, when the input signal SNR is −6 dB, the optimal range of the penalty factor is (3000, 4000) and the optimal range of the penalty factor when the input signal SNR is 1 dB is (1000, 2000)).

### 4.2. Implementation of VMD

Through the analysis and discussion of the VMD parameters above, it can be seen that the parameters in the VMD method have different degrees of influence on the decomposition effect of the input signal. The noise tolerance parameter γ has no decisive influence on the decomposition result, so it can generally be determined according to the preference in actual signal processing. Among them, when the noise margin is 0, the output signal will have a large energy loss; however, the loss of energy will not affect the EEG characteristic frequency in the signal, so for the processing of EEG signals the choice of noise margin does not substantially affect the decomposition results of VMD, thus affecting the removal of artifacts in EEG. The setting of the initial center frequency of another parameter has a certain influence on the decomposition result. However, for EEG, due to their characteristics, the advantages and disadvantages of the VMD decomposition effect under the three initial center frequency setting methods are obvious, so only for this paper the initial center frequency parameter can be directly set to a fixed method. The number of IMFs not only has a non-negligible impact on the VMD itself, but also has an impact on other parameters. Therefore, most of the literature optimizes the VMD parameters for the optimization of the K value of the number of decompositions. From the convergence conditions in the VMD method, it can be known that the VMD decomposition takes the frequency domain changes of the IMF components in the adjacent two cycles as the iterative constraints to determine the iterative stop of the decomposition. In view of this, this paper takes the numerical change of the center frequency between the modal components of the same order in two adjacent decomposition processes as the judgment condition of the preset scale parameter K to determine its optimal value. A K value selection algorithm based on an invalid center frequency is proposed. The specific steps of the method are:
(1)Initialize K value, K=2, and determine the range of K value [2, 10] by analyzing the decomposition results of a large number of original signals and relevant references [25];(2)Perform VMD decomposition to obtain K IMF components and the center frequency of each order signal component $\omega_{K,i}\left(i=1,2,\ldots ,K\right)$, i represents the order;(3)Let K=K+1, and perform VMD decomposition again to obtain K+1 IMF components and the center frequency of each order signal component $\omega_{K+1,j}\left(j=1,2,\ldots ,K+1\right)$;(4)Calculate the judgment accuracy $\varepsilon_{K,a}\left(a=1,2,\ldots ,K\right)$ of the center frequency of each signal component under the same order a with different K values according to the following formula:(18)$$\varepsilon_{K,a}=\frac{\omega_{K,a}}{\omega_{K+1,a}}$$


In the formula, $\varepsilon_{K,a}$ represents the judgment accuracy of the a-order signal component when the number of VMD decomposition modes is K, $\omega_{K,a}$ represents the center frequency of the a.-order signal component when the number of VMD decomposition modes is K, and $\omega_{K+1,a}$ represents the center frequency of the a-order signal component when the number of VMD decomposition modes is K+1;
(5)Determine the size of the judgment accuracy $\varepsilon_{K,a}$ and the accuracy thresholds $\theta_{1}$ and $\theta_{2}$ (the values are 1 and 1.2 after a lot of experiments). If $\varepsilon_{K,a}\geq 1.2$ or $\varepsilon_{K,a}\leq 1$, then it is determined that $\varepsilon_{K,a}$ is an invalid center frequency and vice versa is valid;(6)The K value to which the first invalid center frequency is identified (there may be invalid center frequencies under multiple K values) is the selected K value.


Taking the above simulation signal as an example, the specific process of selecting the optimal K value is as follows:

First calculate the center frequency of K in the range of [2, 10], as shown in Table 4.

According to the formula in step 4 of the method, the determination accuracy of each IMF component when K takes different values is calculated, respectively, and the calculation results are shown in Table 5.

The invalid center frequency is determined according to the threshold determination criterion in method step 5, and it can be seen that the K value corresponding to the first invalid center frequency is 3, so K=3 is determined.

## 5. Discussion

In recent years, the research on single-channel EEG artifact removal has become more and more significant, and there are few studies on the removal of multiple artifacts. Therefore, we propose the VMD-SOBI method to solve the problem of multiple artifacts in a single channel and verify its superiority in solving the single-channel multi-artifact problem by comparing it with the existing EEMD-SOBI method. In addition, it is worth noting the parameter selection problem of VMD. We set up multiple sets of simulation experiments for multiple parameters and analyzed the best choice of each parameter. Among them, the number of modals K is the most important parameter. Here, we propose a method to automatically optimize the K value based on the invalid center frequency, which is more accurate and less time-consuming than some automatic optimization algorithms such as PSO and WAO. In general, this paper proposes a method combining VMD after optimizing the K value and SOBI for the problem of single-channel EEG signal removal of multiple artifacts, which provides a more effective way for biomedical signal processing.

## Figures and Tables

**Figure 1 sensors-22-06698-f001:**
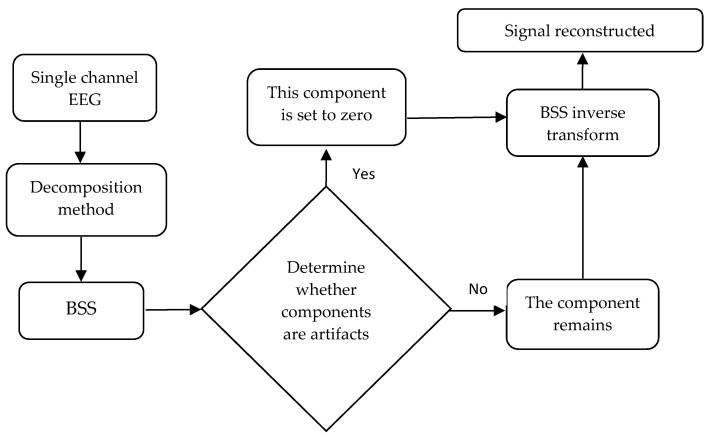
The flow chart of single-channel EEG signal artifact removal.

**Figure 2 sensors-22-06698-f002:**
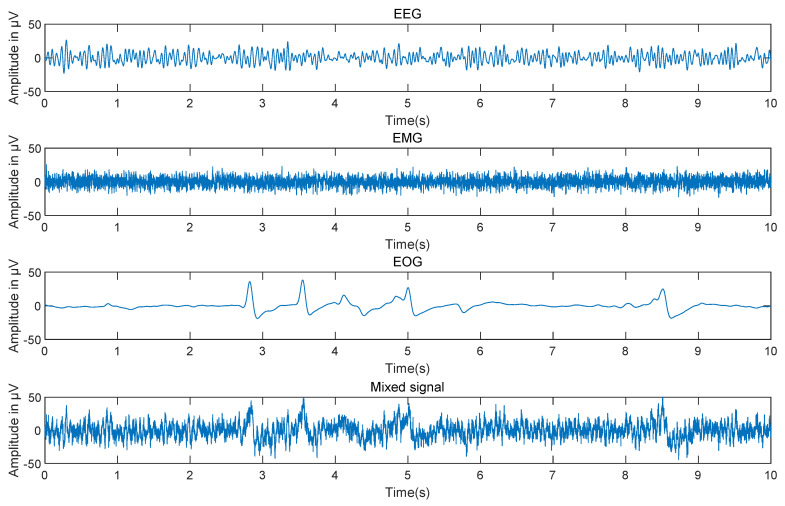
Time domain diagram of simulation signal.

**Figure 3 sensors-22-06698-f003:**
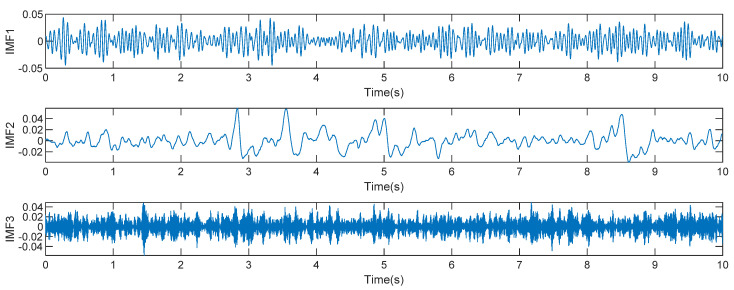
Original artifact containing signal and signal components after VMD-SOBI method.

**Figure 4 sensors-22-06698-f004:**
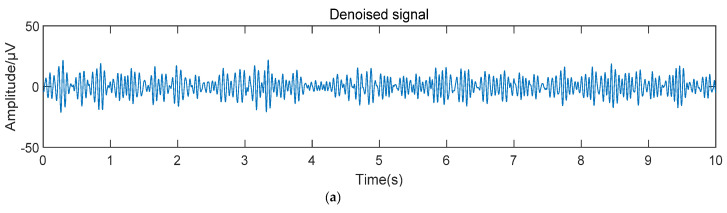
EEG signals after artifact removal by different methods. (**a**) The reconstructed signal after artifact removal by this method. (**b**) The reconstructed signal after artifact removal by EEMD-SOBI method.

**Figure 5 sensors-22-06698-f005:**
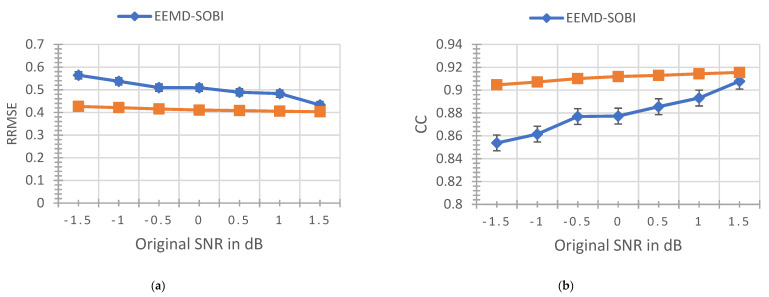
For simulated signals with different signal-to-noise ratios, the statistical results of CC and RMSE of denoising effects of different algorithms. (**a**) Four methods of RRMSE value statistics. (**b**) Four methods of CC value statistics.

**Figure 6 sensors-22-06698-f006:**
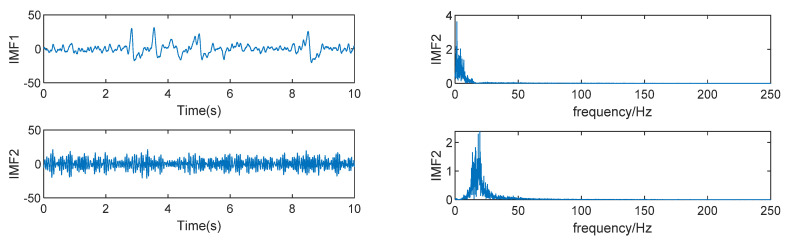
VMD decomposition result of K=2.

**Figure 7 sensors-22-06698-f007:**
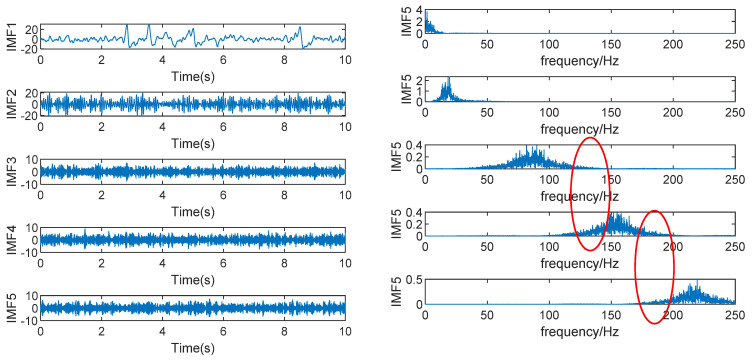
VMD decomposition result of K=5.

**Figure 8 sensors-22-06698-f008:**
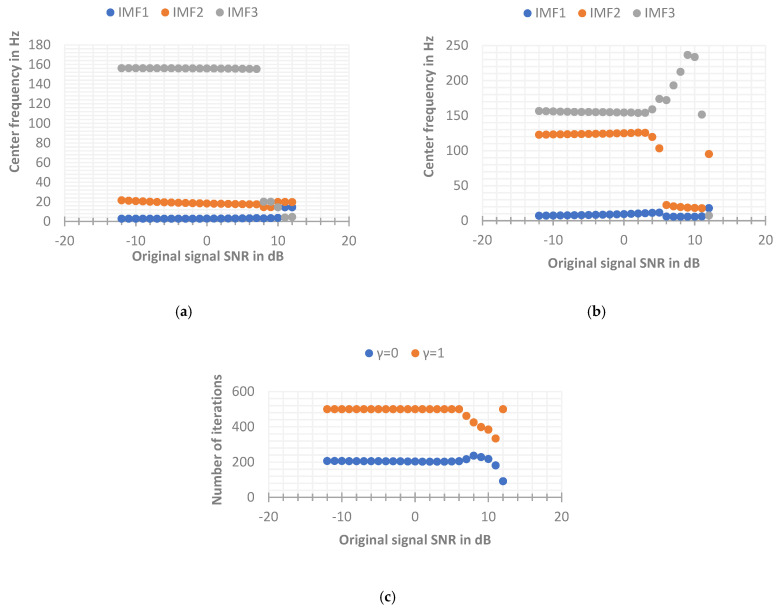
The final convergence center frequency and convergence times of the simulated signal with the initial center frequency is set to 0. (**a**) The components at γ=0 eventually converge to the central frequency. (**b**) The components at γ=1 eventually converge to the central frequency. (**c**) The number of iterations with different values of γ.

**Figure 9 sensors-22-06698-f009:**
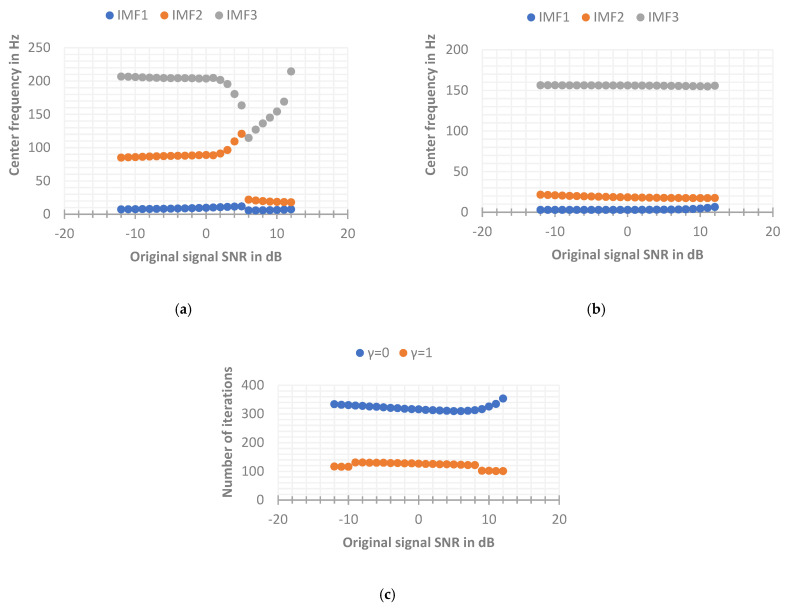
The final convergence center frequency and convergence times of the simulated signal with the initial center frequency set to fixed. (**a**) The components at γ=1 eventually converge to the central frequency. (**b**) The components at γ=0 eventually converge to the central frequency. (**c**) The number of iterations with different values of γ.

**Figure 10 sensors-22-06698-f010:**
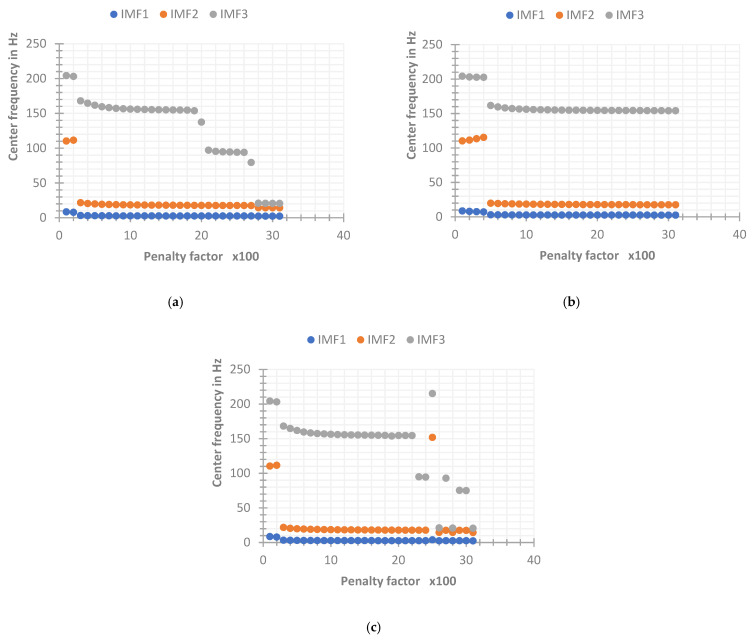
Convergence frequency of different initial center frequency setting methods under −6 dB. (**a**) SNR = −6 dB, the initial center frequency is set to zero. (**b**) SNR = −6 dB, the initial center frequency is set to fixed. (**c**) SNR = −6 dB, the initial center frequency is set to random.

**Figure 11 sensors-22-06698-f011:**
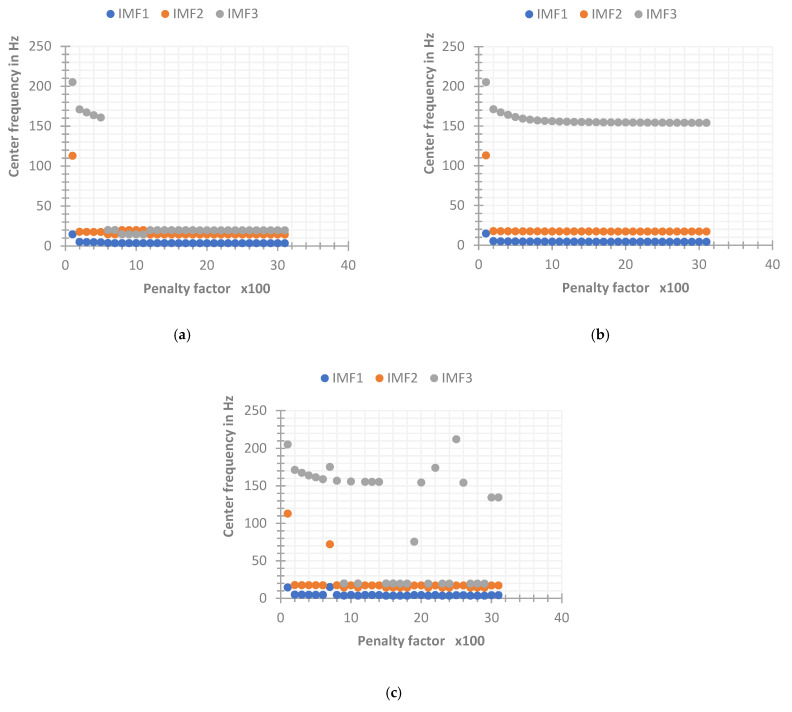
Convergence frequency of different initial center frequency setting methods under 6 dB. (**a**) SNR = 6 dB, the initial center frequency is set to zero. (**b**) SNR = 6 dB, the initial center frequency is set to fixed. (**c**) SNR = 6 dB, the initial center frequency is set to random.

**Figure 12 sensors-22-06698-f012:**
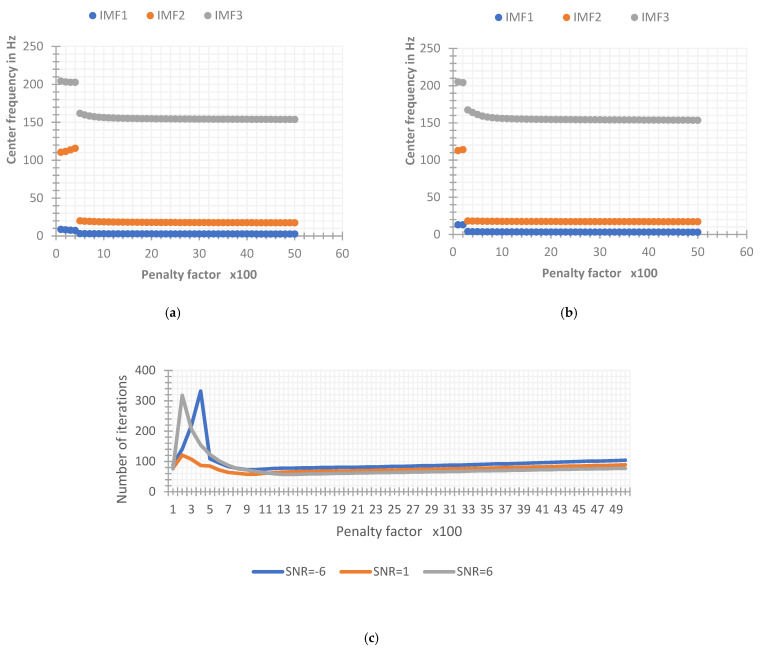
The final convergence frequency and number of iterations of the simulated signal. (**a**) SNR = −6 dB. (**b**) SNR = 1 dB. (**c**) The number of iterations with different SNRs.

**Figure 13 sensors-22-06698-f013:**
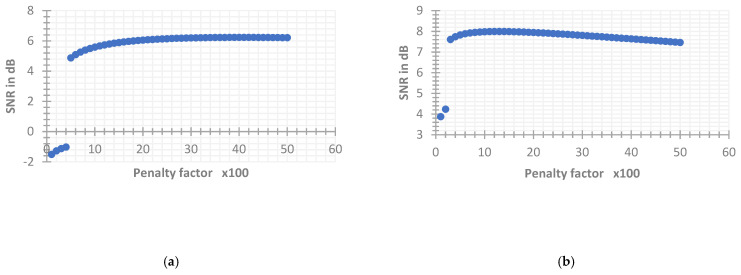
Reconstructed SNR under different original SNR. (**a**) Reconstructed SNR at −6 dB. (**b**) Reconstructed SNR at 1 dB.

**Table 1 sensors-22-06698-t001:** Fuzzy entropy of different signal components.

IC	IC1	IC2	IC3
FE	$1.7\times 10^{-3}$	$2\times 10^{-4}$	$8.84\times 10^{-2}$

**Table 2 sensors-22-06698-t002:** Performance indicators under different methods.

Method	η	σ	γ
EEMD-SOBI	6.2524	0.4865	0.8915
VMD-SOBI	7.8232	0.4052	0.9143

**Table 3 sensors-22-06698-t003:** Frequency range of each source signal.

Source Signal	Frequency Range
EOG	0–5 Hz
EEG	10–50 Hz
EMG	80–250 Hz

**Table 4 sensors-22-06698-t004:** The central frequency at different values of *K*.

K	Center Frequency in Hz									
2	0.0062	0.0356								
3	0.0061	0.0355	0.3122							
4	0.0061	0.0355	0.2406	0.3831						
5	0.0061	0.0353	0.1729	0.3072	0.4327					
6	0.0055	0.0296	0.0419	0.1847	0.3103	0.4337				
7	0.0054	0.0293	0.0413	0.1473	0.2506	0.3480	0.4448			
8	0.0054	0.0292	0.0410	0.1375	0.2293	0.3061	0.3796	0.4615		
9	0.0054	0.0288	0.0404	0.1097	0.1796	0.2483	0.3165	0.3894	0.4665	
10	0.0053	0.0288	0.0403	0.1067	0.1737	0.2390	0.3030	0.3617	0.4231	0.4779

**Table 5 sensors-22-06698-t005:** The judgment accuracy of each order under different *K* values.

K	Judgment Accuracy								
2	1.0037	1.0014							
3	1.0046	1.0019	1.2976						
4	1.0088	1.0042	1.3910	1.2469					
5	1.1065	1.1915	4.1299	1.6630	1.3943				
6	1.0092	1.0113	1.0152	1.2540	1.2381	1.2464			
7	1.0036	1.0043	1.0055	1.0710	1.0932	1.1369	1.1717		
8	1.0096	1.0122	1.0157	1.2540	1.2764	1.2324	1.1993	1.1852	
9	1.0014	1.0017	1.0022	1.0284	1.0341	1.0389	1.0445	1.0766	1.1026

## Data Availability

Links to public datasets used in the experiments: https://www.bbci.de/competition/iv/ (accessed on 4 September 2021). The code used in the experiment: https://github.com/LCR997/V-sobi (accessed on 4 September 2021).

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
