# Peer review of "Remove Artifacts from a Single-Channel EEG Based on VMD and SOBI"

_sensors, 2022, doi:10.3390/s22176698_

Round 1

Reviewer 1 Report

In this study, the authors develop a method for removing artifacts from single-channel EEG signals. The method uses variational modal decomposition (VMD) to decompose a single channel into many modes (simulating having multiple channels). The method then uses second-order blind identification (SOBI), a variation of independent component analysis (ICA), to decompose the newly created multi-channel data into candidate noise components. Finally, a subset of components is selected for removal, based on fuzzy entropy (signal complexity), and the original single-channel EEG is pruned, similar to ICA (inverse transform after blind source separation). My overall assessment of the work is positive, but I think the paper could strongly benefit from changes to the overall presentation. Specific comments are listed below, roughly in their order of importance.

Comment 1: In my opinion, the main takeaway for this paper is that VMD-SOBI (new method) is better than EEMD-SOBI (old method), as shown in Figure 13. Unfortunately, this isn't shown to the reader until the very end. Although I appreciate the effort that went into Section 2, I think the average reader could lose focus of what is most important. I highly recommend rearranging the content of the paper, making use of supplemental sections, to better guide readers to the main takeaway. I think the majority of the content in sections 2.2-2.5 can be quickly summarized, leaving only the most important (high-level) ideas in the main body, and then supplemental sections can be referenced for more detail. Some people may not immediately understand the mathematics in 2.2 and stop reading the paper even though there is still something for them to gain from your new approach. Similarly, in 2.3, the actual content itself is good and the effort was worthwhile; however, to appeal to a broader audience, I recommend moving these details to a supplementary section.  In my opinion, the sooner readers see there is real, quantitative promise to your newly proposed method (VMD-SOBI) over an existing method (EEMD-SOBI), the sooner they will be interested in diving into the details. 

Comment 2: Many readers will not be able to recreate the algorithm, even given the information in Section 2. Please provide source code (written functional scripts) so others can more easily implement the novel method in their own research and to verify its performance.

Comment 3: Currently figure captions contain very little information. Rather than duplicating information, I would recommend moving text from the body of the manuscript and placing it in the figure captions. I think you can appeal to a broader audience if the figures are self-sufficient and don’t require reading of the main text.

Comment 4: Please double-check the mathematical equations for errors. Variable f from equation (2) is undefined. Superscript notation in equation (4) is undefined. At line 154, I believe it should be “until n = K” not “until k = K”. Also, k and K are used without being initialized.  

Comment 5: The Discussion should more clearly state the conclusions.

Comment 6: Please comment on the real-world speed of this approach as compared to other cleaning algorithms (processing time in hours/minutes/seconds).

Comment 7: At line 22 (in the Abstract), it says the proposed method was compared to other existing “methods” but it was only compared to a single method, not plural.

Comment 8: Please consider using another symbol at line 185 instead of Lambda. This parameter, which is a simple scalar multiplier that controls the amount of noise in the signal, could be confused with the Lagrangian multiplication operation you introduced in 140.

Comment 9: The sentence from lines 200-204 is not clear.

Comment 10: please replace “sampling point” in all figures with the time (in seconds).

Comment 11: At line 53, please re-word the phase “affect its effect”

Comment 12: At line 85, “IMF”, intrinsic mode function, is not yet defined

Reviewer 2 Report

The Authors provided a new method for artirfacts removal from EEG signal which is a combination of VMD and SOBI methods. 

The paper is structured very well, the Authors provided quite a thorough explanation of the rationale of their method and how it improves the existing methods (to support this they provided a relavant comparison showing adventages of their approach).  Even if the improvement is not spectacular, every little helps. So from this point of view the paper ticks the box about contributing to the knowledge and as such is worth further processing.

I have some reservations to the editing standard due to the fact that some figures clearly are not formatted / alligned properly with the text (but I am not sure whether this is the view which only I can see or it is rather this is in issue needing the Authors' attention. This should be closely looked at and possibly  improved (I don't need to revisit the paper, only I want to make sure that the Authors will double check this).  Other than that - no serious reservations over the presentation standard. Also, use of English is fine so the paper is easy to follow. 

Round 2

Reviewer 1 Report

I am satisfied with the authors' edits to the manuscript. The comments below are minor.

1) Please check the label for the bottommost plot in Fig 2.  I believe this is the mixture of all 3 sources ("EEG+EMG+EOG") but it is labeled as "EOG".

2) Please write the x label as "Time (s)" rather than "t/s"

3) In Fig 5., it would be helpful to note to readers (who are possibly skimming the paper) that lower RRMSE is better and higher CC is better.
